# Debunking misleading graphs effectively: How vocationally educated young adults perceive graphs

Winnifred Wijnker[1], Peter Burger[2], Ionica Smeets[3], Sanne Willems[4]*

1 Research groups Qualitative Journalism in Digital Transition and Human Experience & Media Design, Utrecht University of Applied Sciences, Utrecht, the Netherlands, 2 Journalism & New Media, Leiden University, Leiden, the Netherlands, 3 Science Communication & Society, Leiden University, Leiden, the Netherlands, 4 Department of Epidemiology and Data Science, Amsterdam UMC, location University of Amsterdam, Amsterdam, the Netherlands

* s.j.w.willems@amsterdamumc.nl

## Abstract

Misleading graphs can give readers a distorted view of the underlying data. We want to know how to most effectively correct misleading graphs and if it matters whether a correction uses the full-design of the original or a clean design with all embellishment and colors removed. We focus on vocationally educated young adults, a group that is vulnerable to misinformation and has so far been underrepresented in research. We use a mixed-method approach with a qualitative think-aloud task (n = 10, data collected in April 2023) and a quantitative survey (n = 130, data collected between July and October 2023). The think-aloud task showed that vocational students use a combination of calculating and estimating to process graphs, which contradicts existing literature, and that their perception is heavily influenced by context. The survey showed that graph corrections work in reducing misleading effects and also have a learning effect such that students are less misled by new misleading graphs of the same type. There was no difference between full-design and clean design corrections. These results imply that vocationally educated young adults can benefit from seeing corrections of misleading graphs.

## 1. Introduction

Either produced intentionally or accidentally with best intentions, incomplete, partially true, and outright fake news pollutes the public domain and it is up to the reader to draw accurate conclusions from sometimes ambiguous information in texts, videos, photographs, illustrations, and graphs. Fact-checkers and researchers are looking for ways to protect readers and viewers from misinformation. In this study we focus specifically on how this can be done for misleading graphs. Warning readers against possible deceit and activating graph reading skills is assumed to stimulate conscious

**Data availability statement:** The study design, research questions, hypotheses, and analyses were preregistered at AsPredicted. org (https://aspredicted.org/8Z5_YSL). The collected quantitative data and transcripts of the interviews are stored and publicly available at Open Science Framework (https://osf.io/gcmf2/overview).

**Funding:** This research was supported by Leiden University Fund (http://www.luf.nl, LUF Lustrum Grant 2020, W20719-1-LLS).

**Competing interests:** The authors have declared that no competing interests exist.

analysis [1,2]. Some even argue for inoculation of readers by exposing them to misinformation techniques [3]. Many of these strategies are found to be quite effective; but they require readers who understand the danger of misinformation and are willing to invest time and effort in learning new skills. A technique to debunk misinformation more directly is to offer a fact-check. Repeated exposure to fact-checks may make readers less susceptible to misinformation in the long run [4,5].

Effective fact-checks for written articles provide arguments rather than a simple retraction and do not repeat the misleading message [6–9]. In line with these findings, our previous experiment showed that presenting the correct graph next to the misleading graph was the most effective debunking strategy for a general audience, tested in a clean experimental set-up [10]. In the study presented here, we take our strategy to a more realistic approach and aim to improve the design for an audience that might benefit most from it: vocationally educated young adults.

Young adults are constantly gaining new responsibilities and have a growing independence that requires them to increasingly make informed decisions – possibly based on graphically displayed information. It is a group that is very active on social media where visually displayed information is easily and often shared, without professional evaluation. We focus on vocationally educated participants because participants with a practice-oriented education are presumed to be more easily misled than those with a research-oriented education [10], and are underrepresented in graph reading studies and in research in general [11].

This study addresses the question of how vocationally educated young adults interpret misleading graphs and what design options can best guide the process of seeing through the deceit.

## 2. Theoretical framework

### 2.1. How we read graphs

For presenting related numbers, graphs can enable a quick overview [12]. They allow the reader to look up specific data points, to understand relationships between the data, and to make predictions beyond the data [13].

A graphical presentation of numbers provides easy access to the data, but that does not guarantee a quick and easy read and may require thorough studying for full comprehension [14]. Beside the complexity of the data, how much effort it takes to read a graph is also dependent on the reader's motivation and graph reading skills.

The motivation to engage with a specific graph in turn is dependent on the reader's interest in the data that the graph presents. Processing graphs starts rather effortlessly based on associations with learned general regularities, but then requires more effort when the reader moves on to intentional rule-based inferences [15]. If interest is low, the reader might stop engaging after the first phase of graph reading; the phase of looking [14]. In this phase, readers glance over the graph, comparing heights of bars, depicted areas, or portions of pies, or estimate the steepness of lines. This first phase results in a preliminary conclusion regarding the topic, such as "The trend is going down" or "Category A takes up a bigger piece than category B" [16]. Highly interested readers might continue with the second phase of graph

reading; the phase of more conscious reading [14]. In this phase, readers go beyond the imaging and start reading titles, subtitles, category names, legends, and numbers. At this point, readers can distinguish actual data points and compare them to each other [2].

Readers that take the effort to enter the second phase of graph reading might come to the same conclusions as did the less motivated readers, but there are two factors that influence this similarity or difference greatly: the design of the graph [17], and the aforementioned graph reading skills of the reader [13].

A graph's design plays a crucial role in the first phase of graph reading, the phase of looking [13]. Well-designed graphs should lead low and highly interested readers to draw roughly the same conclusions on the data and their relationships, i.e., they should be perceptually accurate such that the first impression matches with the true message. However, there are many ways in which graphs may display a distorted image of the data, due to intentional or naïve poor design choices [18–20]. For example, in well-designed bar graphs, pictorial area charts, and pie charts, the area of the elements depicting different categories (respectively bars, icons, and pie parts) should be proportional to the values they represent. In bar graphs, an accurate presentation of the values requires that the vertical axis starts at zero because cut-off vertical axes shorten the bars and make the differences between the bars appear bigger. In pictorial area charts, an accurate representation requires keeping in mind that the icon is scaled in two dimensions to maintain good proportions. That way, an icon that is twice the height of its neighbor, has an area that is four times bigger and thus should also represent a number that is four times bigger. In pie charts, an accurate representation requires that the pie is portrayed in 2D, because a 3D representation makes front parts appear bigger than back and side parts. See Fig 1 for examples of poor graph design.

Sticking to common design conventions can help to prevent perceptual errors [21]. Cut-off vertical axes, wrongly scaled icons, and 3D pies complicate a proper comparison between the data. We regard graphs with design elements that obstruct a proper understanding of the data as misleading.

Beside accuracy of the depicted data, design choices concern the overall layout of the graph. There are leads from graph design practice to believe that a clean design that minimizes use of ink and unnecessary design elements would be more effective, because the lack of possibly distracting embellishing elements enables the reader to focus on the data itself [20]. However, to our knowledge this has not been proven in research. Comparative studies on accurate graph reading have shown that 'unnecessary' imaging in graphs does not obstruct data perception as compared to clean designs and may even improve recall of the graph and its message [22–24], but effects may vary per task and context [25,26].

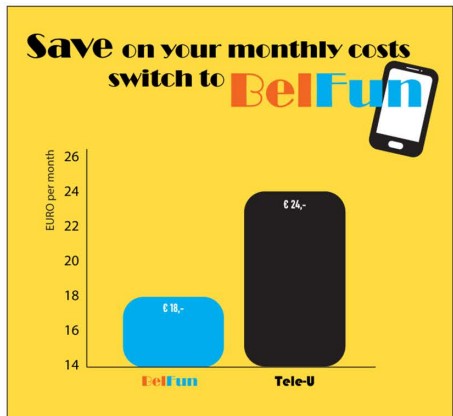 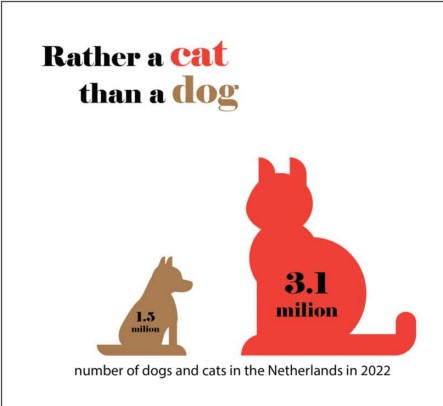 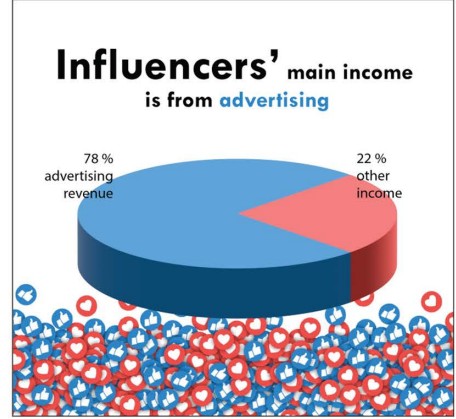

**Fig 1. Examples of poor graph design.** Examples of a poorly designed bar graph (left), pictorial area chart (center), and pie chart (right). All graphs presented in this paper were created for this study with fictional data using Adobe Illustrator 2023 and 2024.

Psychology research confirms this effect on recall, showing that images prove to be a powerful tool for memorability [27–30]. Additionally, it can be expected that an attractive design may trigger readers' interest.

### 2.2. How to overcome misleading graphs

Highly interested readers that are inclined to fully engage with the graph might partially counter misleading graphs in the second phase of graph reading when studying the data in more detail. However, the damage done in the first phase is very unlikely to become undone in the second phase due to the strength of the sticky initial image and the persistent preliminary conclusions of the reader [2].

People that are used to reading graphs and dealing with data might enter the second phase more easily than those who are not [2]. People that are less familiar with graphs can be expected to be less inclined to enter the second phase of graph reading, because when they do they have to invest more effort to find the relevant information. A distorted effort-reward balance is known to make interest drop [31], making graph novices more vulnerable to misleading graphs. How well people are able to deal with graph reading is referred to as their graph reading skills or graph literacy and can be measured with a test such as the Graph Literacy Scale [32] (see S1 Fig in the Supporting information for the items included in the test).

In this study we used a mixed methods approach by combining a thinking-aloud task to gather qualitative data with an online survey for quantitative data. The qualitative part of this study addresses the explorative research question (RQ1) *How do vocationally educated young adults interpret misleading graphs and their corrections?* The main question for the quantitative part is: (RQ2) *What is the most effective design (clean or full) of corrected misleading graphs for vocationally educated young adults?*

## 3. Methods

The study design, research questions, hypotheses, and analyses were preregistered at AsPredicted.org (https://aspredicted.org/8Z5_YSL) and have been approved by the Psychology Research Ethics Committee of Leiden University, the Netherlands, with reference number 2023-01-25-S.J.W. Willems-V3-4381 for the interviews and number 2023-07-03-S.J.W. Willems-V2-4876 for the online survey. The quantitative data was collected using Qualtrics' online survey software. The survey instruments and questions, collected quantitative data, transcripts of the interviews, and code for data analysis are stored and publicly available at Open Science Framework (https://osf.io/gcmf2/overview).

### 3.1. Study design

In collaboration with six vocational education teachers (in the Netherlands: MBO), we set up a class meeting with six of their student groups (5–26 students per group) about graphs, how to read graphs, and about misinformation. The students in the classroom filled in a questionnaire that started with an informed consent. Participants that gave consent were linked to the questionnaire of which the data was saved; the ones that did not give consent were linked to a copy of the questionnaire of which the data was deleted. At the end of the questionnaire, the students were debriefed about the purpose of study and informed that all graphs in the questionnaire showed fictional data. The participants that gave consent to use their data were given the option to withdraw their permission at the end of the questionnaire. The data collected in the classrooms we found to be not suitable for further analysis (see below for further details) and was treated as a pilot study to further refine the questionnaire and set-up.

From each class one or two students were invited to voluntarily fill in the questionnaire while thinking-aloud in an interview instead, outside the classroom. Before the interview started, the students were informed and asked for consent similarly to the students that simultaneously filled in the questionnaire in the classroom, but with additional information on the recording of the interview. The interviewed participants received the same debriefing and possibility to opt out as the classroom participants.

In the class meeting, the questionnaire functioned as a starter to familiarize the students with bar graphs, pictorial area graphs, and pie charts, and to introduce misleading designs. The in-class questionnaire and the interviews were followed by a guest lecture on misleading graphs from the researchers, so that this project also had an educational purpose. The interviews and classroom visits took place in April 2023.

While collecting data in the classrooms, we got the impression not all participants were filling in the questionnaire seriously. In the interviews, some graphs or questions were confusing to the participants and needed explanation by the researchers (see S1 File for details). Therefore, we decided not to use the classroom survey data and set up a new data collection. We improved the questionnaire with feedback from the interviews and set out an online survey for quantitative data collection, through voluntary participation via KiesKompas.nl (https://www.kieskompas.nl/en/). KiesKompas collected data between July and October 2023. For the online survey, the same informed consent, debriefing and opt-out was used as in the classroom.

### 3.2. Participants

Our main research question for the quantitative part (RQ2) focused on the difference between the correction effect of two correction types for misleading graphs. To reach 80% power for a direct comparison of the mean correction effect of the two correction types, 64 participants were required per group to detect a medium effect size ($d = 0.5$), i.e., 128 participants in total (independent samples two-sided $t$-test, alpha = .05). KiesKompas recruited 206 participants that at the time of data collection were enrolled in vocational education programs or graduated the year before and were aged 16–25 years. Exclusion criteria were not completing the questionnaire (76 participants), and finishing it unrealistically quick, i.e., within two minutes (no participants). Participants that did not fully complete the demographics part or the test for graph reading skills (4 participants) were not excluded. Our final sample consisted of 130 participants. For the qualitative part we interviewed 10 students (5 female, 5 male; aged 16–22) enrolled in vocational education programs.

### 3.3. Instruments

#### 3.3.1. Questionnaire.
We set up a questionnaire that was used in interviews as a thinking-aloud task and for an online survey. First, the participants were asked to provide demographics. Next, participants were asked to evaluate the difference between the data of two categories presented in the graphs on a visual analogue scale (VAS) ranging from "very small" to "very big" which was transcoded into values between 0 (very small) to 100 (very big). We deliberately did not ask participants to find and report specific data points, to avoid interfering with activation of their graph reading skills. At the baseline measure, the participants were presented the first randomized series of 12 graphs, containing four bar graphs (two accurate, two misleading), four pictorial area graphs (idem), and four pie charts (idem). The participants were randomly assigned to either group A or B: Group A was presented accurate versions of the misleading graphs that were presented to group B, and vice versa (see Fig 2).

Next, the participants received the correction "treatments" to evaluate corrections of the misleading graphs they already evaluated. The participants were randomly presented one of the two design options for corrections (clean or full-design). After the corrections, the participants evaluated a mixed series of new misleading and new accurate graphs. Again, group A was presented accurate versions of the misleading graphs that were presented to group B, and vice versa.

Finally, the participants fulfilled a four-item test to measure their graph reading skills (Short Graph Literacy scale, [32]). Completing the survey took approximately 15–20 minutes.

#### 3.3.2. Thinking-aloud task.
The participants invited for an interview were asked to fill in the questionnaire while thinking aloud. For the task, the participants were instructed to indicate what they were looking at, to articulate their thoughts, and to motivate their evaluation of each graph. The participants were stimulated to keep talking while processing the graphs according to a predefined protocol (ask: what are you looking at/what goes on in your mind?). When things were unclear, the researcher would explain. The interview was recorded and transcribed for further analysis. The interviews took approximately 20–30 minutes.

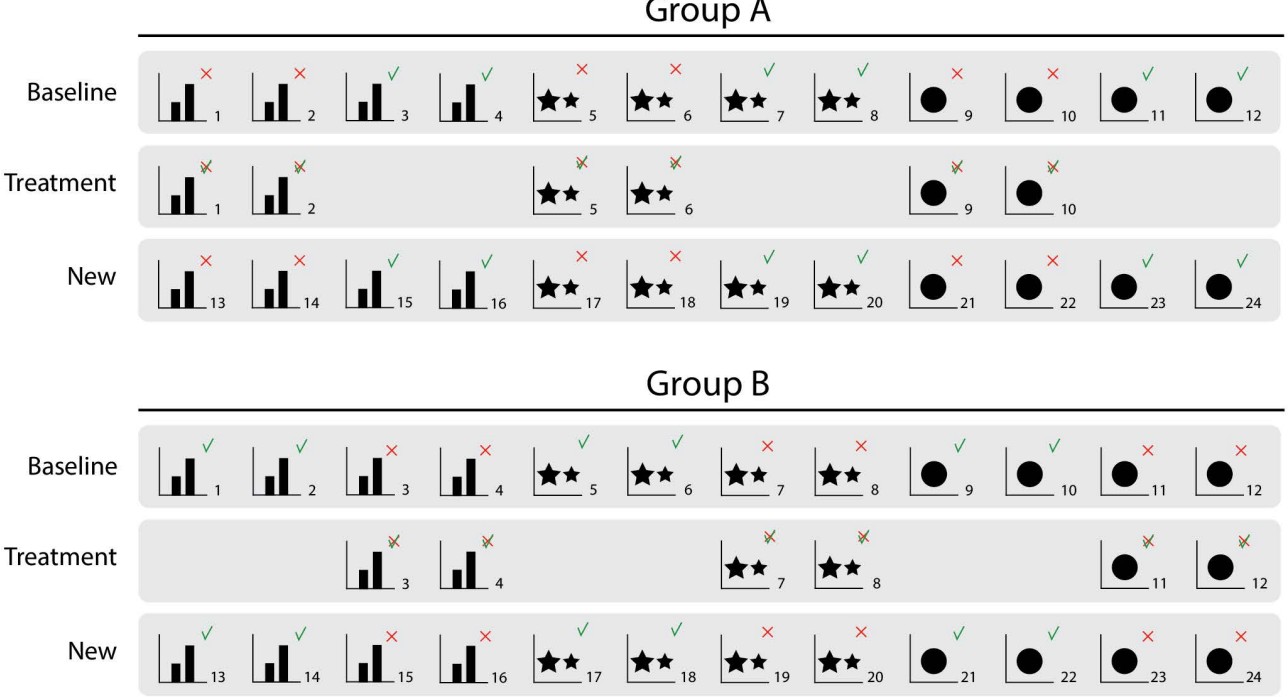

**Fig 2. Study set-up.** Series of bar graphs (e.g., graph 1), pictorial area charts (e.g., 5), and pie charts (e.g., 9) per stage (see far left). The graphs were randomized within each series per participant. The series consisted of accurate graphs (with a green check mark), misleading graphs (with a red cross), and corrected versions of the misleading graphs (marked with a cross and a check mark). The treatment consisted of a clean or full-design correction. Participants were randomly assigned to a group and a correction design.

### 3.3.3. Graphs and corrections.

In the questionnaire 24 unique graphs were used. We included bar charts, pictorial area graphs and pie charts (eight of each graph type). Each graph displayed data on two categories or times. The graphs provided fictional information about topics that were chosen to be relevant to young adults, such as fees for phone subscriptions, social media use, or the popularity of Netflix series. The data was fictional but close to reality and the design was aligned to the colorful popular designs used by online media (see Fig 3).

We balanced the position of the smaller bar (left or right), the smaller pictogram in area charts (left or right), and the smaller proportion of the pie (left or right) over the complete set of graphs. The bars roughly showed a difference of 25% between the bars, the pictorial area charts showed a 50% difference, and the pies roughly showed a distribution of 25 vs. 75%. For bar graphs and pictorial area graphs, the evaluation question was formulated as "How do you evaluate the difference between [category/time 1] and [category/time 2]?". For pie charts the question was formulated as "How do you evaluate the proportion of [category/time 1 or 2]?". The selection of the category (1 or 2) that was mentioned in the evaluation question for pie charts was matched to the emphasis in the title of the graph.

We created an accurate and a misleading version of each graph. Misleading bar graphs had truncated vertical axes, in misleading pictorial area graphs the icons were scaled on height while maintaining the proportions instead of scaled to area, and misleading pie charts were presented in 3D designs. For each misleading graph we create a correction in two different designs. One design mimics the original and is referred to as "full-design", and the other minimizes use of color and embellishment and is referred to as "clean design" (see Fig 4).

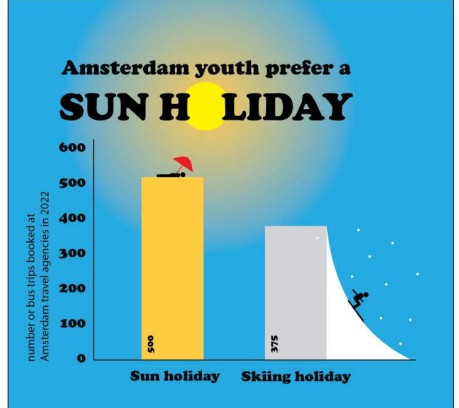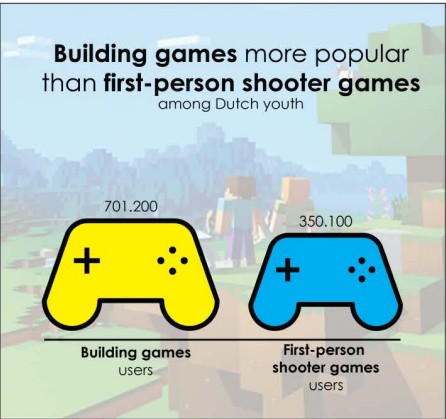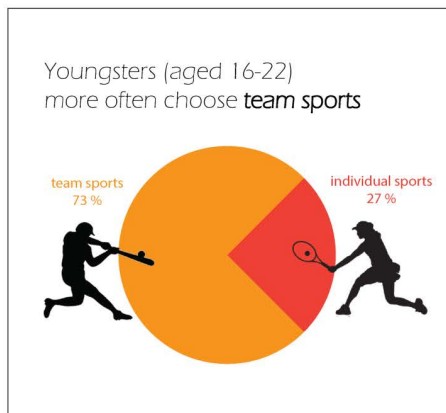

**Fig 3. Examples of accurate graphs.** Examples of an accurate bar graph (left), pictorial area graph (center), and pie chart (right).

### 3.4. Analyses

**3.4.1. Qualitative interview data.** The interview data were coded in two phases, firstly by descriptive coding and secondly by axial coding, following Saldaña [33]. After the first round of analysis by the first author, the last author recoded 20% of the data. The outcomes and differences were discussed between the researchers to complement the code book and to optimize code definitions. Then, most relevant codes were selected, and the data were coded again (see S1 Table for the final codebook). No statistical analyses were performed on these data.

**3.4.2. Quantitative survey data (preregistered and alterations).** With the survey, we aimed to study the general effect of showing corrections of misleading graphs, and whether there are differences between the two designs (clean or full-design). The research question was answered in steps via a set of hypotheses, where the answer to each hypothesis is first studied by visualizing the raw data and then confirming any observations using several linear mixed effect models. This is a deviation from our pre-registered plan to fit one large linear mixed effects model, as this full model would include so many variables and interactions that interpreting it would become too complicated. As we noticed differences between the graph types in the manipulation check, an additional deviation from the preregistration is that we immediately include graph type as a variable in the main analyses, which was originally planned to be a separate exploratory step.

All data analyses were done in R version 4.2.2 and the R code is available via this project's Open Science Framework page.

**3.4.2.1. Manipulation check: were graphs indeed misleading?** To check whether the misleading graphs that we designed were indeed misleading, we preregistered to test the following hypothesis (Fig 5a visualizes this hypothesis, i.e., what graphs as displayed in Fig 2 were involved in this manipulation check).

- H1: Misleading graphs lead to a higher evaluation of the difference in data than the accurate graphs on the same context in the other group.

We preregistered to test this hypothesis for all contexts individually with independent samples one-sided *t*-tests (with Holm-Bonferroni correction for multiple testing). And, if a graph was not found to be misleading, we would consider excluding this context from further analyses. Slightly deviating from the pre-registration, we only performed these analyses on the contexts that were shown as the first set of graphs in the survey, i.e., before any corrections were shown, to avoid any influences from the corrections.

As testing for all context individually reduced the power of the tests, we also performed a second set of independent samples one-sided *t*-tests, with one test for each graph type instead of for each context. In this way we only need to correct for multiple testing of three conditions instead of twelve. More details are given in the results.

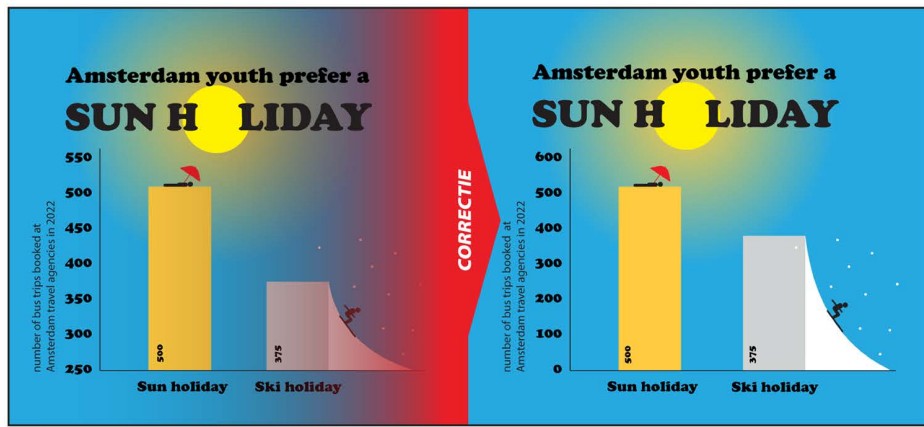

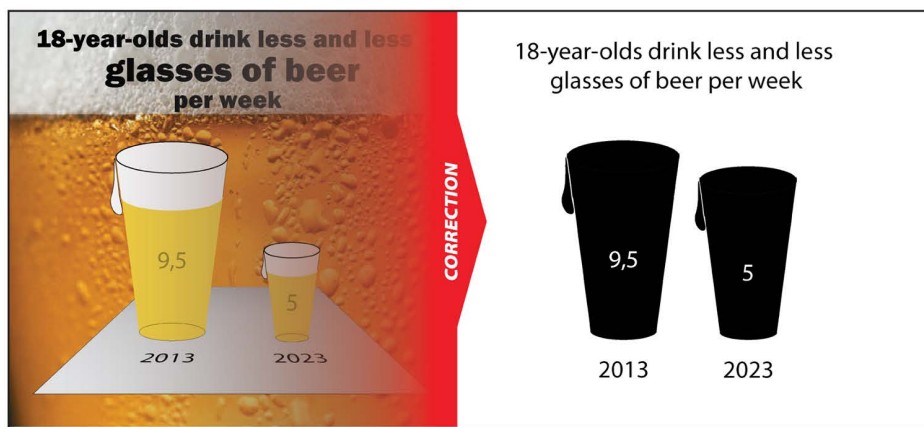

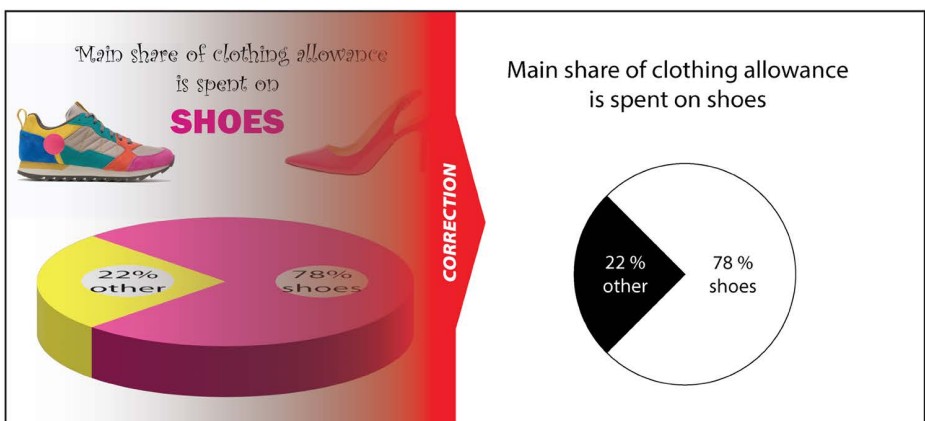

**Fig 4. Examples of misleading graphs.** Example of a misleading bar graph correction (top) in full-design, and a misleading pictorial area graph (center) and pie chart (bottom) correction in clean design.

**3.4.2.2. Direct effect of showing a correction (H2-H3):** To study the immediate debunking effect, we test whether showing corrections has a direct effect on the evaluation of the misleading graphs, and whether there is any difference between the two designs (clean vs. full-design). We defined the following hypotheses (visualized in Fig 5b) to answer these questions.

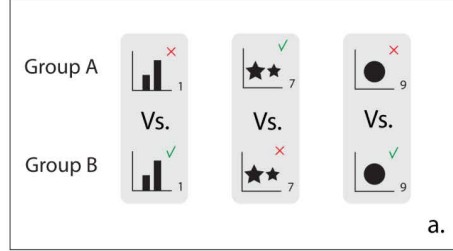 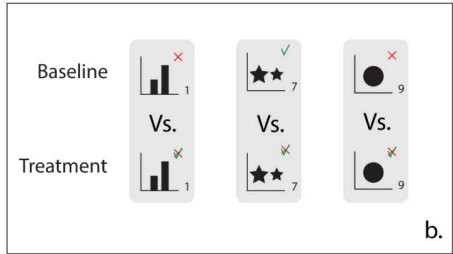 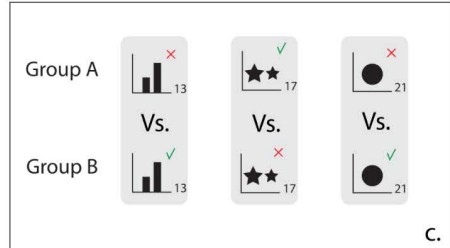

**Fig 5. Visualizations of the hypotheses. a.** For H1 we compared evaluations of misleading and accurate graphs at baseline; **b.** For H2 we compared misleading graphs at baseline and accurate graphs at treatment. For H3 we included the full/clean design condition in this comparison; **c.** For H4 we compared new misleading and accurate graphs (stage new). For H5 we included the full/clean design condition in this comparison.

- H2: Corrections of misleading graphs directly lead to lower perceived differences compared to the initial perceptions of the same misleading graphs.

- H3: Correction effects at correction (H2) differ per correction type (clean/full-design).

Hence, H2 and H3 focus on the comparison of the evaluations of the misleading graphs at baseline and their corrections at treatment (see Fig 5b), i.e. the direct effect of showing a correction.

The linear mixed effects model for H2 predicts the evaluation score and includes the misleadingness (misleading vs. corrected), graph type (bar/pictorial area/pie charts) and their interaction as fixed effect, and graph contexts and participants as random effects to accommodate for any random differences between the contexts and participants.

To answer H3, the evaluations of the two correction types at treatment are directly compared with a linear mixed model that includes the correction types (clean design vs. full-design), graph type and their interaction as fixed effect, and graph contexts and participants as random effects.

**3.4.2.3. Learning effect of showing a correction (H4-H5):** Secondly, we wonder whether showing corrections of misleading graphs has a learning effect, i.e., whether the debunking effect extends to new misleading graphs, and if the design of the correction has an effect on this. The following hypotheses were defined (and visualized in Fig 5c) to answer these research questions.

- H4: Corrections of misleading graphs directly lead to lower perceived differences in new misleading graphs compared to perceptions of previously evaluated misleading graphs.

- H5: Correction effects for new graphs directly after intervention (H4) differ per correction type (clean/full-design).

Hence, H4 and H5 focus on the comparison of the evaluations of the misleading and accurate versions of the new graphs that are shown after the correction treatment (see Fig 5c).

The linear mixed effects model to test H4 includes fixed effects for the misleadingness of the graphs, graph type, and their interaction, and again graph contexts and participants as random effects. A similar model is fitted to compare the evaluations of the accurate and misleading graphs at baseline, so that we can directly compare the misleadingness of misleading graphs at baseline and after the corrections.

To test the effect of the design of the correction (H5), the mixed effects model fitted on the new graphs is extended to include the correction type as a fixed effect, together with its interaction with graph type and misleadingness.

**3.4.2.4. Exploratory analyses:** As an additional exploratory part in this study, we preregistered to extend the above-mentioned linear mixed effects model with demographic variables to check for individual differences affecting the correction effects. As our main interest is in the effect of graph literacy, models were only extended with this fixed effect.

The results of these exploratory analyses are described along with the results of the above-mentioned hypotheses, focusing on those that showed significant effects.

## 4. Results

### 4.1. Qualitative results from the interviews

**4.1.1. Demographics and graph reading skills.** Ten students (5 female, 5 male; aged 16−22) were interviewed while performing a thinking-aloud task. Their educational level ranges from secondary vocational education levels MBO-2–4 (MBO1: entrance training; MBO2: basic vocational training; MBO3: vocational training; MBO4: specialist training). Their scores on the Short Graph Literacy Scale range from 1 to 3 (see Table 1).

**4.1.2. Graph reading and evaluation processes.** Almost without exception, students started off by reading the title of the graph (see Quote 1), then the categories and the values indicated, and then the evaluation question with the VAS. It is hard to tell from the audio recordings at what point the participants scanned the imaging of the graph. If there were remarks about the heights of the bars, the size of the areas or embellishments, they mostly came only after this process or were only made upon request of the researcher to describe what the participant was looking at.

> Quote 1 (participant #9): "*Most Dutch youngsters wait to have sex until the age of eighteen, 75% waits, 25% does not wait. How do you evaluate the proportion of youngsters who have sex before the age of eighteen?*"

Students used two strategies when they started evaluating the difference between the presented categories: 1) calculating the difference between the given values, or 2) estimating the size difference between the icons, bars, or pie parts. Most students used a combination of the two, starting off by calculating and then referring to the image and how the size difference is in line with the values they represent. This deviates from the theory on graph reading as a process first of looking (estimating the difference based on the imaging), and then of reading (gathering and subtracting the relevant values to calculate the difference). After calculation and estimation, students would state a first judgement on the found difference (big or small). The strategy was consistent in most students over the different graphs and graph types.

**4.1.3. Full-design vs. clean design preferences.** In the interviews all students were shown the clean design correction of the full design graphs to enable evaluation of the different design options. Some students indicated they appreciated the clarity of the clean designs of the corrections because the embellishment in the original full design graphs distracted them a lot from the data (Quote 2). Others found the clean design corrections boring (Quote 3). Of students that

**Table 1. Demographics and graph literacy scores (GLS) of the interviewed participants.**

| Participant # | Gender | Age (M = 18,7 SD = 2,0) | MBO level (M = 3,4 SD = 0,8) | GLS (M = 2,1 SD = 0,7) |
|---|---|---|---|---|
| 1 | Male | 18 | 4 | 2 |
| 2 | Male | 17 | 4 | 2 |
| 3 | Female | 22 | 2 | 1 |
| 4 | Female | 17 | 3 | 3 |
| 5 | Female | 16 | 3 | 3 |
| 6 | Male | 19 | 4 | 3 |
| 7 | Male | 19 | 4 | 1 |
| 8 | Male | 21 | 4 | 2 |
| 9 | Female | 21 | 4 | 2 |
| 10 | Female | 17 | 2 | 2 |

expressed a preference for either full or clean designs, the preference was consistent for all graphs within individuals. We did not find an overall preference for either clean or full-design.

> Quote 2 (pp# 10): *"Most dating apps violate privacy legislation rules, what a distracting image this is. All these hearts are standing in the way. You hardly notice the text here down below, I would easily miss that."*

> Quote 3 (pp# 5): *"Well, this one is quite a lot bigger and in 3D, and everything, you can really see depth in it, and that one is just super, well, boring actually, literally black and white."*

**4.1.4. Relevance of contexts.** Often the first judgement was motivated by how the presented figures did or did not align with the participant's own knowledge or experiences regarding the topic. Figures that aligned with their preconceptions generally received a moderate evaluation, whereas figures that did not align received more extreme evaluations (very small/very big). In response to the researcher's request to elaborate on their evaluation, students would for example declare how they recognized the displayed differences in their own lives (Quote 4), or how they did not believe the presented values could be real (Quote 5). Some remarked being concerned about the severity of the figures (Quote 6), or agreeing with the figures (Quote 7).

> Quote 4 (pp# 2): *"Well, I don't think the difference is that big, because people can get more cats than dogs."*

> Quote 5 (pp# 7): *"Most Dutch young adults wait until the age of eighteen to have sex, oh no they don't. […] We live in a society that is very transparent, so I wouldn't think that people would wait, no."*

> Quote 6 (pp# 9): *"Well, you can see 75 percent that violate privacy legislation and 25 percent that do not, while it should actually be the other way around. Because, well, you agree, you know, with privacy legislation, so this is already too much. That is really not done!"*

> Quote 7 (pp# 3): *"Actually, I think it's okay. Everybody has the right to decide how many animals they want to have."*

**4.1.5 Deceit awareness.** At baseline measures, from the three types of graph deceit with which the students were confronted, the cut-off y-axis in misleading bar graphs was mentioned most often. Many students simply took the fact that the y-axis did not start at zero as a given and did not draw any consequence from it (coded as Level 1 deceit recognition; Quote 8). Only a few recognized this as a misleading element and referred to how the difference between the values did not match the difference between the heights of the bars (coded as Level 2 deceit recognition; Quote 9). From the pictorial area charts in the baseline measures, it was mentioned only a few times that the areas did not match the values they were to represent. However, participants that had spotted the incongruence almost without exception classified this mismatch as being misleading (Quote 10). The 3D effect in pie charts was not mentioned as a misleading element at baseline.

> Quote 8 (pp# 6): *"And then it [the vertical axis] starts at 250, goes up to 550, with steps of 50."*

> Quote 9 (pp# 8): *"Well, at first glance it would seem like a lot, but if you look at the numbers, well, there isn't much of a difference."*

> Quote 10 (pp# 5): *"Because the cat is so big, you know, and the dog is so small... 'What do you think of the difference between the number of dogs and cats?' Quite large actually, but I think it is also because the picture is very large, you know, because the cat is also drawn very large here. That's why I think "Wow", but actually yes 1.5 and 3.1, I think it's not that big."*

In the new misleading graphs, after the participants were presented with corrections of the misleading graphs, almost all cut-off y-axes in bar graphs were spotted and regarded as misleading (Quote 11). The misleading pictorial area charts

were sometimes recognized and found to be misleading (Quote 12), and the 3D effect in pie charts was noticed but not often regarded as being misleading. Participants that recognized mismatches between values and value representations at baseline measures (Level 2) most often would describe how this design choice made the difference between the data look bigger but did not regard it to be intentionally deceiving. After the correction their choice of wording changed to categorizing it as misleading and they would express a feeling of triumph about seeing through the deceit (Quote 13 and 14).

Quote 11 (pp# 3): *"That image is actually, this is much too long, I think, because there are not that many."*

Quote 12 (pp# 2): *"[…] it looks like a quarter, but it is half. At first sight it seems nice, but…"*

Quote 13 (pp# 10): *"It doesn't start at zero, now it's starting to show."*

Quote 14 (pp# 5): *"Yes it is quite misleading, you have to keep thinking, and keep looking at the numbers. But yes, it influences you a lot."*

### 4.2. Quantitative results from the online survey

**4.2.1. Demographics and graph literacy scores.** The participant characteristics are described in Table 2. Most participants are in the highest MBO level (MBO4). The distribution of graph literacy scale scores is quite high in comparison to other studies with representative samples (e.g., [32]).

**4.2.2. Manipulation check – Were the graphs indeed misleading? (H1).** To get a first impression of whether graphs were indeed misleading, we plot the distributions of the evaluations on the VAS scale for each context separately, and for each graph type to spot any structural differences between the bar graphs, pictorial area graphs and pie charts, see Fig 6. For easy comparison, we added dots indicating the group means.

In Fig 6 we see that in all contexts except for "Dating Apps", the mean evaluation on the VAS of the accurate graphs is lower than the evaluation of the misleading graphs. This suggests that generally the graphs were indeed misleading, i.e., the difference between the classes shown in the accurate graphs were experienced to be smaller compared to their difference in the misleading graphs. However, the differences between these means are quite small for most contexts, except

**Table 2. Demographic information of participants, including their graph literacy sum scores.**

| Characteristics | | Total $n = 130$ |
|---|---|---|
| **Age, *M* (*SD*)** | | 20.56 (2.26) |
| **Gender, %** | Man | 64 (49.2) |
| | Woman | 62 (47.7) |
| | Other | 4 (3.1) |
| **MBO level, (%)** | level 1 | 0 (0) |
| | level 2 | 5 (3.8) |
| | level 3 | 18 (13.8) |
| | level 4 | 102 (78.5) |
| | *missing* | 5 (3.8) |
| **Graph literacy sum scores (%)** | 0 | 7 (5.4) |
| | 1 | 19 (14.6) |
| | 2 | 32 (24.6) |
| | 3 | 47 (36.2) |
| | 4 | 21 (16.2) |
| | *missing* | 4 (3.1) |

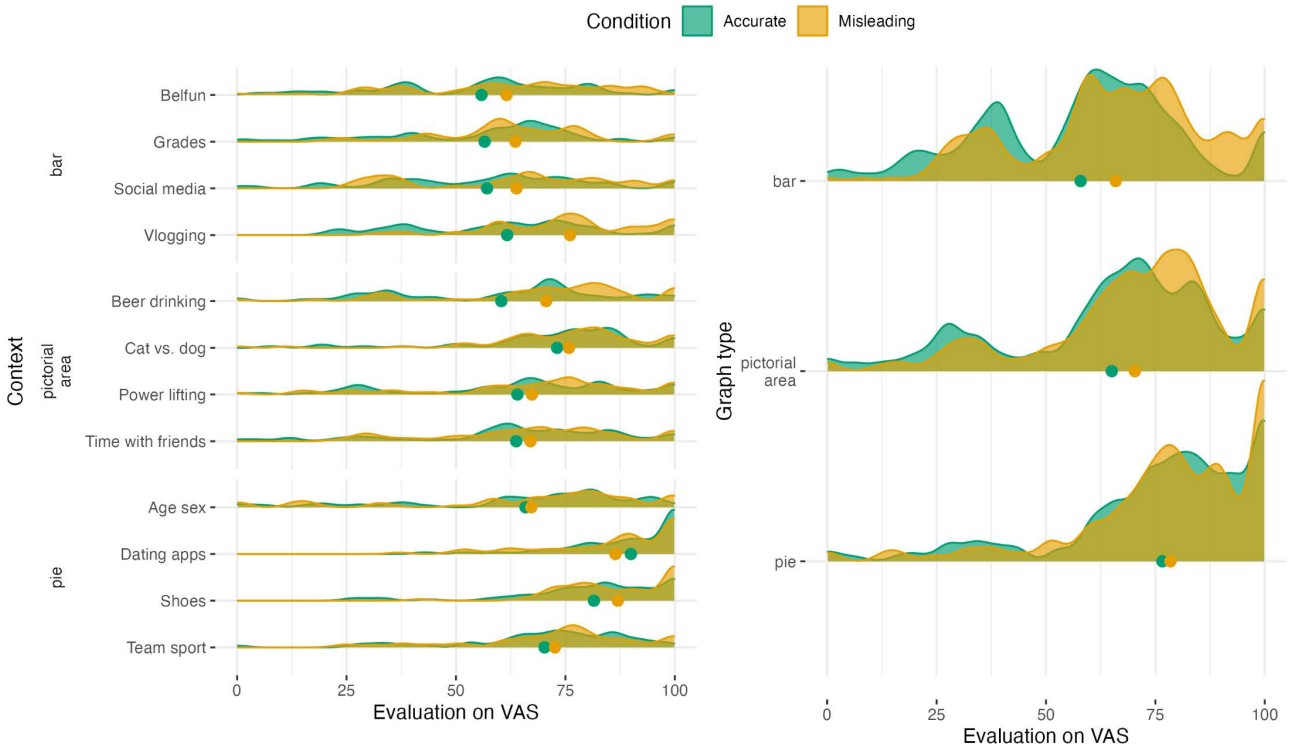

**Fig 6. Density plots showing the distribution of the evaluation scores of the accurate and misleading graphs on the VAS scale at baseline, separate for each context (left) or together for each graph type (right).** Density plots are smoothed versions of histograms that represent the raw distribution of the answers of the participants in the various conditions. Smoothed tails are cut off at the answer range endpoints 0 and 100 and the dots indicate the mean values per condition.

for "Vlogging", which is the only contexts for which the *t*-test (corrected for multiple testing) shows a significant difference (see Table 1 in S2 File). This result may mean that the graphs were generally not misleading, or that we do not have enough power to show significant differences in the evaluations of the misleading and accurate graphs per context.

The lack of power is caused by the multiple-testing correction that is needed since we are running *t*-tests for 12 different contexts. With the multiple-testing correction, the differences between the two conditions need to be quite extreme to be significant after the correction. To increase the power we instead ran the manipulation check per graph type (bar/pictorial area/pie charts), as visualized in Fig 6 (right). Here, we see a difference between the evaluations of misleading and accurate bar and pictorial area graphs, but not for the pie chart where the dots representing the means almost completely overlap. This observation is confirmed by the *t*-tests (with multiple testing correction for the three graph types), which show significant differences for the bar and pictorial area graphs (see Table 2 in S2 File). Hence, from these observations we can conclude that indeed, generally, the misleading bar and pictorial area graphs were misleading, but some contexts resulted in stronger differences.

For exploratory purposes, we ran additional models to study the effect of graph literacy on manipulation at baseline, but found no significant effect for the graph literacy score (Table 3 in S2 File). Hence we see no evidence that a higher graph literacy score protects against misleading graphs.

**4.2.3 The direct effect of showing a correction (H2-H3).** As all participants were shown a correction for the misleading graphs that they saw in the first round of the survey, we can study the correction effect by comparing their evaluations either per context or per graph type, see Fig 7. These density plots show that for all graph types the

evaluations for the corrections (yellow and blue) reduce the initial evaluation of the misleading graphs (orange), but the effect differs per context (Fig 7 – left). Interestingly, this drop on the VAS scale is even observed for pie charts, which were originally not very misleading (see results H1).

The mixed effects model confirms that, generally, showing a correction will reduce the evaluation by 10.00 points on the VAS (see Table 1, Model A in S3 File), but that there is no significant interaction between correction effects and the three graph types. This is in line with the observations in Fig 7 that correcting a misleading graph reduced the initial evaluation, and shows that there is no significant difference of this reduction between graph types.

No significant influence of graph literacy was found on the debunking effect of corrected graphs (see Table 1, Model B in S3 File).

When comparing the effects of the two correction designs in Fig 7, we observe that for each context and graph type the effect is generally the strongest for the correction with a full-design (yellow points are generally to the left of the blue points). However, the difference in the evaluations of these two designs is not significant (Table 2 in S3 File).

**4.2.4 Learning effect of showing a correction (H4-H5).** An additional interest is whether the effect of showing a correction has an effect on the participants' evaluation of future misleading graphs, i.e., do they learn anything from the correction?

The differences in evaluations between the new misleading graphs (shown to one group of participants) and accurate graphs (shown to the other group) are visualized in Fig 8. We observe that generally misleading graphs still get higher evaluations than accurate graphs (except for the pie chart on job success). This is confirmed by the mixed effects model

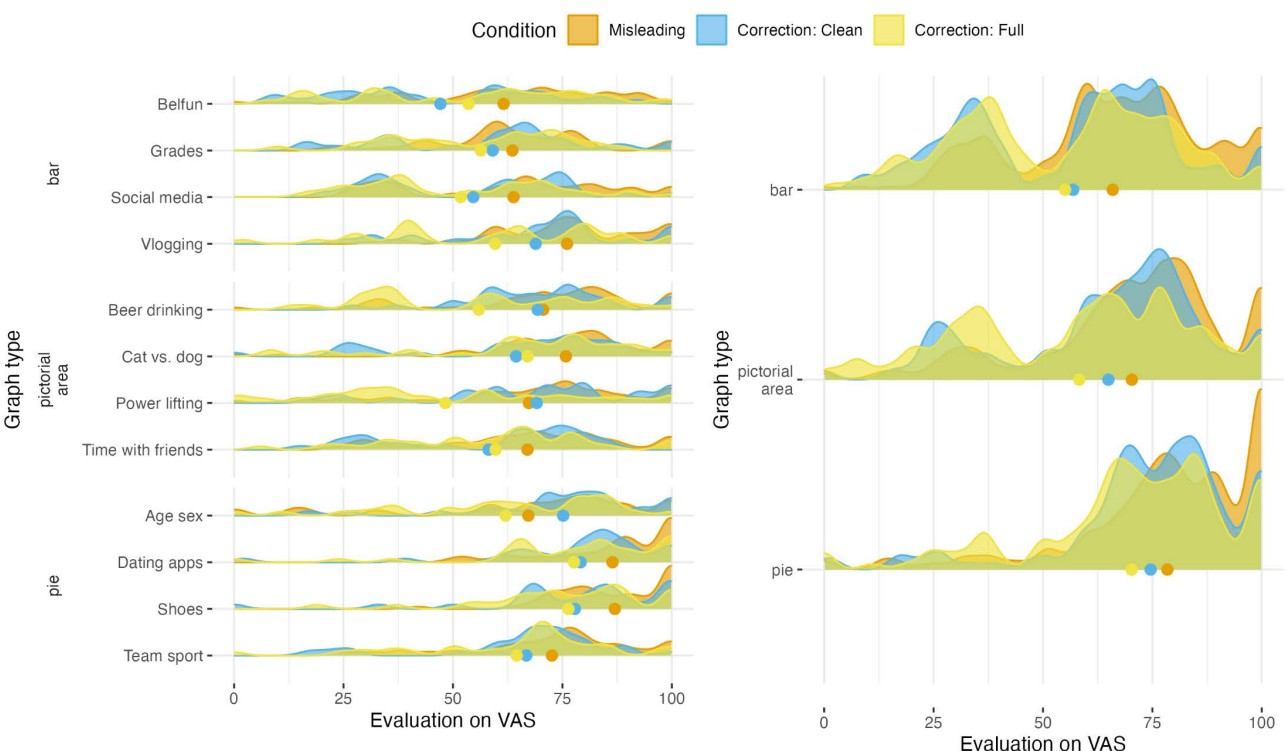

**Fig 7. Density plots showing the distribution of the evaluation scores on the VAS scale, comparing the misleading graphs at baseline and their corrected version (in either clean design or full-design).** Density plots are smoothed versions of histograms that represent the raw distribution of the answers of the participants in the various conditions. Smoothed tails are cut off at the answer range endpoints 0 and 100 and the dots indicate the mean values per condition.

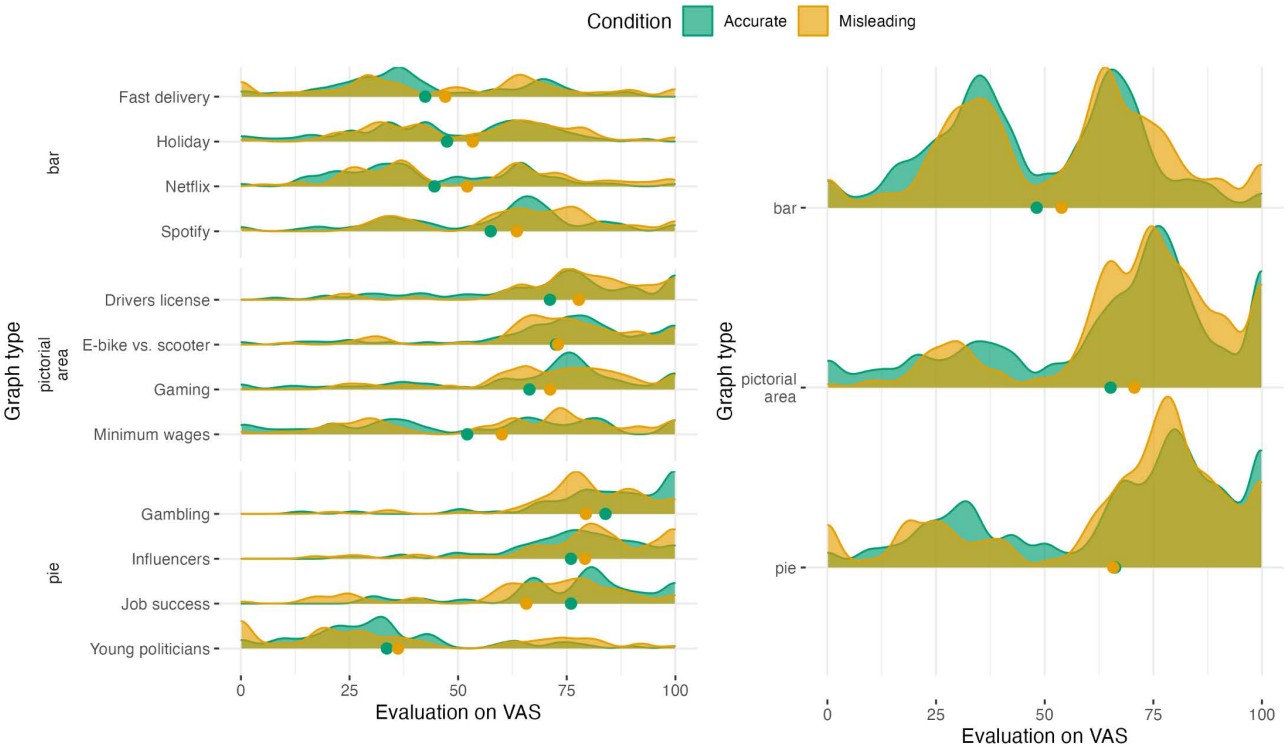

**Fig 8. Density plots showing the distribution of the evaluation scores of the new accurate and misleading graphs on the VAS scale after seeing corrections, separate for each context (left) or together for each graph type (right).** Density plots are smoothed versions of histograms that represent the raw distribution of the answers of the participants in the various conditions. Smoothed tails are cut off at the answer range endpoints 0 and 100 and the dots indicate the mean values per condition.

which shows a significant difference of 6.01 across all graph types (see Table 1, New in S4 File). At baseline, this difference was slightly higher, namely 8.43 (Table 1, Baseline in S4 File). Hence, there seems to be a small learning effect reducing the misleadingness of the graphs by 2.42 points on the VAS scale. However, as the direct effect of showing a correction was 10.00 points, the learning effect is much smaller. Hence, showing corrections does not completely eliminate the misleading effect of new misleading graphs. Additionally, we found no significant difference in learning effect between the two correction designs (see Fig and Table 2 in S4 File).

Finally, we observed that especially for the pie charts and bar graphs, the evaluations for both the new accurate and misleading graphs are generally much lower than the evaluations at baseline (compare means and distributions in Fig 6 with Fig 8). However, when corrected for context, individuals, and graph type in a mixed effects model (see details and model results in Appendix SF), this drop is not significant. Hence, there is no evidence that the corrections affect the general evaluation of graphs. This drop is possibly due to the differences in contexts of the graphs.

## 5. Discussion and conclusion

Visual media like graphs aid readers to quickly oversee large sets of sometimes complex data and make messages more memorable. However, intentionally or by accident, not all graphs have a perceptually accurate design, resulting in misleading data representations. Hasty or less motivated readers and graph novices easily overlook design violations. The initial image of misleading graphs displaying exaggerated differences is hard to counter, and drawing accurate conclusions requires readers' attention and effort. A direct graph correction presenting an accurate alternative has proven to

be an effective debunking tool for all readers. Young adults with vocational-education backgrounds are potentially more vulnerable to misleading graphs. In this study, we aimed to better understand how this population reads graphs, and to optimize the correction design for them.

Our findings show that offering a correction is more important than considering its design. Both clean and full-design corrections showed a decrease in the difference evaluation and made participants more attentive to misleading graphs. Preferences for either design differ per person and both have their pros and cons.

## 5.1 Graph reading and evaluation processes

Most participants of the thinking-aloud task followed the same strategy: reading the title, reading the numbers with the graph, then reading the evaluation question before coming to a first judgement. This is not in line with the theories on graph reading processes that were the basis of our set-up, discerning first a stage of looking, and then a stage of reading [2,14,16]. The strategy of our participants might have been the result of the instruction to think aloud, assuming that reading out words and numbers is easier than formulating what is seen. From our data we cannot be sure about their exact strategy; the participants probably did look before reading – if only to locate the words and numbers.

It was hard to tell at what point the participants scanned the design of the graph and what they were looking at. Despite our encouragements to keep talking, only a few made remarks about where their eyes were taking them and what they saw. Hence, recording eye-tracking data could be a valuable addition to the thinking-aloud task in future studies [34]. However, we valued collecting data in a real educational setting over collecting more data in a lab setting where eye trackers can easily be set-up, and therefore choose not to include eye tracking in our study.

## 5.2 Relevance of contexts

In the interviews, we noticed that besides evaluating the difference of the presented data, participants would evaluate the severity of the topic, the credibility of the data, or the degree to which the data aligned with their own experience. For example, the graph on violation of privacy legislation in dating apps evoked strong emotions in many participants. This aligns with the survey data, showing the highest mean evaluation score and the biggest peak at 100 ("very big"; see Fig 7).

The results from the survey data show a decrease in the difference evaluation at correction for most corrected graphs as compared to the misleading graphs. We regard this as an implication of the effect of the corrections. However, given that the contexts of the graphs were obviously taken into consideration while evaluating the differences in the data, we should not rule out the possibility that the lower evaluations could also reflect tampered emotions when seeing the graph for the second time at Correction.

Although the VAS did not display any numbers, we also noticed that some participants were matching the position of the slider to the data ratio in pie charts. This could explain the diverging results for the pie chart on young politicians, which was the only pie chart asking to evaluate the smaller part (see Fig 8).

## 5.3 Deceit awareness

Violation of graph conventions was spotted most often in bar graphs (the cut-off y-axis). Participants that mentioned it would sometimes discuss how this design choice made the data differences appear different from what they calculated, but not always, and most often without classifying it as misleading design. After correction, many more participants spotted the cut-off y-axis and their choice of wording changed to categorizing it as misleading and seeing through the deceit was often accompanied by a feeling of triumph. However, there were many occasions in which the participants took the cut-off y-axis simply as a given. The 3D effect was never mentioned as a misleading effect. Pictorial area charts were often not even recognized as charts, but rather discussed as images depicting what the numbers were about.

As bar and line charts are the most common graph types, frequently seeing them might make it easier to spot misleading design. On the contrary, accurate reading of pie charts and pictorial area charts may be obstructed without notice due to unfamiliarity. Since in daily life, we encounter all sorts of graphs with all kinds of embellishments and unconventional design choices [35], this may make readers less prepared for the many misleading graphs.

### 5.4 Constraints on generality

In our study we put in extra effort to reach a specific group of participants that takes up the biggest share in Dutch society: vocationally educated young adults.

However, this is not an easy-to-find group of participants. To recruit the participants, we personally got into contact with vocational education teachers. They gladly invited us into their classrooms, expressing their appreciation for "academia" reaching out to their students. In their experience, their students are often neglected in research and forgotten when educational material is developed. This critique is backed in academic debates [11].

We experienced first-hand how difficult it can be to collect data from this group. In the classroom, the students often did not take the questionnaire seriously. The participants were easily distracted or clicked through the survey with little attention. We believe to have seen a confirmation of our doubts in the data we collected in the classrooms, showing quite some peaks at the 'convenient evaluation values' zero, 50 and 100 (see Appendix SC). For the online data collection, KiesKompas was struggling to reach the required number of participants. A possible explanation is that this group is not too keen on or interested in participating in research studies. More research is needed for more clarity on this issue.

The teachers confirmed that research is not something their students care for. They rather have some practical tips they can immediately use in their own lives. Personal contact is the key to their world and minds. This might be the reason why the interviews turned out to have generated so much valuable information. In a one-on-one conversation, the students were very open and honest in their answers. So, yes, this was not an easy ride. But we do believe it was well-worth the effort and encourage researchers to invest in further inclusion of unheard voices in research.

## Supporting information

**S1 Fig. Items of the Short Graph Literacy (SGL) scale.** As developed by Okan, Janssen, Galesic and Waters (2019).
(TIFF)

**S1 File. Comparison of data collected at the MBO schools and online via KiesKompas.** Including S1 File Fig.
(PDF)

**S2 File. Analyses manipulation check – Were the graphs indeed misleading? (H1).** Including S2 File Tables 1–3.
(PDF)

**S3 File. Analyses of direct effect of showing a correction (H2-H3).** Including S3 File Tables 1 and 2.
(PDF)

**S4 File. Models fitted to determine the learning effect of showing a correction.** Including S4 File Tables 1–3, and S4 File Fig.
(PDF)

**S1 Table. Codebook for the thinking-aloud task.**
(PDF)

## Acknowledgments

We want to thank KiesKompas.nl for their effort in reaching our target audience. We thank Maaike Boele and Lonneke Boels for their help in designing and teaching the guest lectures, and Floris Kennis for his contribution to the literature review. We thank Alderic Weijs for helping to establish the collaboration with vocational education teachers Anechina de Jong, Angela Malmberg, Bart Steenbergen, Bernadette Houtdijk, Houda el Abbouti, and Peter Streekstra.

## Author contributions

**Conceptualization:** Winnifred Wijnker, Peter Burger, Ionica Smeets, Sanne Willems.

**Data curation:** Winnifred Wijnker, Sanne Willems.

**Formal analysis:** Winnifred Wijnker, Sanne Willems.

**Funding acquisition:** Peter Burger, Ionica Smeets, Sanne Willems.

**Investigation:** Winnifred Wijnker, Sanne Willems.

**Methodology:** Winnifred Wijnker, Sanne Willems.

**Project administration:** Winnifred Wijnker.

**Resources:** Winnifred Wijnker, Sanne Willems.

**Software:** Winnifred Wijnker, Sanne Willems.

**Supervision:** Ionica Smeets.

**Visualization:** Winnifred Wijnker, Sanne Willems.

**Writing – original draft:** Winnifred Wijnker, Sanne Willems.

**Writing – review & editing:** Peter Burger, Ionica Smeets.

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
