## [Decision Letter · Decision Letter 0]

20 Oct 2024

Dear Dr. Wijnker,

We look forward to receiving your revised manuscript.

Kind regards,

Massimo Stella, PhD

Academic Editor

PLOS ONE

Journal Requirements:

“This research was supported by Leiden University Fund (http://www.luf.nl, LUF Lustrum Grant 2020, W20719-1-LLS).”

3. Please note that your Data Availability Statement is currently missing the direct link to access each database. If your manuscript is accepted for publication, you will be asked to provide these details on a very short timeline. We therefore suggest that you provide this information now, though we will not hold up the peer review process if you are unable.

4. We note that Figures 1 left and 4 top in your submission contain copyrighted images. All PLOS content is published under the Creative Commons Attribution License (CC BY 4.0), which means that the manuscript, images, and Supporting Information files will be freely available online, and any third party is permitted to access, download, copy, distribute, and use these materials in any way, even commercially, with proper attribution. For more information, see our copyright guidelines: http://journals.plos.org/plosone/s/licenses-and-copyright .

a. You may seek permission from the original copyright holder of Figures 1 left and 4 top to publish the content specifically under the CC BY 4.0 license.

Reviewers' comments:

Reviewer's Responses to Questions

**Comments to the Author**

1. Is the manuscript technically sound, and do the data support the conclusions?

Reviewer #1: Yes

Reviewer #2: Yes

Reviewer #3: Partly

2. Has the statistical analysis been performed appropriately and rigorously?

Reviewer #1: Yes

Reviewer #2: Yes

Reviewer #3: No

3. Have the authors made all data underlying the findings in their manuscript fully available?

Reviewer #1: Yes

Reviewer #2: Yes

Reviewer #3: Yes

4. Is the manuscript presented in an intelligible fashion and written in standard English?

Reviewer #1: Yes

Reviewer #2: Yes

Reviewer #3: No

Reviewer #1: This article investigates how vocationally educated students in the Netherlands perceive misleading graphs and how exposure to corrected graphs can improve their graph reading skills and can aid in sensitizing them for being misled less in the future by misleading graphs. The study compares two forms of graph correction: in a clean format (color and extra items removed) and in a full design. The authors use a mixed-method approach with qualitative think-aloud interviews and a quantitative survey. This mixed-method approach appears to be the best choice for this research since vocationally educated students - as mentioned by the researchers - appear to care little for research which complicates quantitative studies with online questionnaires. Indeed, the authors mention that they struggled recruiting enough participants for the online questionnaire and that participants during the thinking-aloud tasks seemed more motivated and “very open and honest in their answers”. Therefore, the qualitative interviews appear to be well suited for vocationally educated people who are still underrepresented in research. This choice was a big asset of the paper which is very well written and is an interesting and original contribution to the existing literature. The relevance of the investigation of vocationally educated adults (an underrepresented group in research) is also clearly highlighted.

In the following I specify a list of minor comments for the revision of the article.

Minor comments:

1. Introduction; paragraph 2. The authors write that the graphs in a previous study were tested “in a clean experimental set-up” and in the current study in a “more realistic setting”. In which ways was the present study a more realistic setting than the previous clean setup? Data was gathered from participants online in an online questionnaire. How was this more realistic? And also the thinking-aloud interviews seem to have taken place in a rather experimental setting. Please elaborate on this with a short example.

2. Theoretical framework; 2.3; paragraph 3. Please specify here what the “clean design” encompasses. It is explained later on in the text (for example in section 3.3.3), but a short elaboration here with an example would be useful.

3. Methods; 3.1; paragraph 1. How big were the student groups on average and how many teachers were involved for each class? If there were six different teachers and one class for each teacher, certain teacher-specific differences might influence the outcome. Furthermore, if the classes were very large this could also affect the outcomes.

4. Methods; 3.1; paragraph 2. How were the students for the thinking-aloud interviews chosen? Careful not to consider them to be representative for the entire class because if they volunteered, their motivation for participation may already be higher than for the other students who were also mentioned not to have taken the questionnaire seriously.

5. Methods; 3.1; paragraph 4. The authors mention that not all participants were taking the questionnaire seriously. In addition to the reasons given by the authors, this might also be because the students seem to have been somewhat forced to participate in the questionnaire. Even if they could refuse their approval for data collection, they were still forced to complete the questionnaire which is another reason why the decision of the authors to discard these data was justified.

6. Methods; 3.2; paragraph 1. It was not clear at first that the student data was discarded completely and new participants independent from the classrooms were recruited. I had the impression at first that a new data collection with the same students was set up. Please clarify this here or even in the section above. This might be cleared up by mentioning the classroom data was used as a sort of piloting and testing of the study.

7. Methods; 3.2; paragraph 1. Please report on the gender and age distribution of the participants in the main text. Also specify how the 10 students for the qualitative interviews were recruited with which inclusion or exclusion criteria.

8. Methods; 3.3.2; paragraph 1. What are the “predefined protocols” mentioned that were used during the thinking-aloud task? Were these the same as the evaluation questions mentioned later?

9. Methods; 3.3.3; paragraph 1. Was the fictional information which was used to create the graphs surprising to the participants or was it following common expectations, i.e. what a majority of people believe or expect to be true? Graphs containing surprising information that contradicts their own expectations might spark higher interest in participants, tempting them to look at the graph more closely and this might aid in them spotting misleading features. Graphs, however, that depict a common belief might not be looked at in that much detail and misleading features in these might be more easily overlooked. Please specify which form of fictional information was used and whether this might have had an impact on the results.

10. Results; Throughout the results section and the Discussion section please specify how many participants mentioned something or used a certain combination instead of just writing “most students” (p.16), “most often” (p. 18), “most participants” (p.25), “many more participants” (p.27), “many occasion” (p.27). Report the actual numbers or percentages in brackets in the text.

11. Results; 4.1.2; paragraph 1. In the first quote it was not clear to me whether the participant was reading the instructions or commenting on the graph. It sounds like reading the instructions so this quote does not seem to contribute any meaning to how the participant was evaluating the graph.

12. Results; 4.1.3; paragraph 1. The authors write: “In the interviews all students were shown the clean design correction”. Does this mean they never saw the full design correction and only the clean design corrections? Or did they see both? But then again the participants were commenting on how boring the clean design was but not the full design correction? Please explain.

13. Discussion; 5.1; paragraph 1. There is an extra open bracket after the references [14][16][2].

14. Discussion; 5.4; paragraph 1. The authors claim that the results found for vocationally educated students are “generalizable to a large group of people” because they take up the biggest share in Dutch society. This fact may make the results more relevant to a large majority, but not more generalizable to the whole. The authors even state below that the participants in their study had much higher graph literacy scores than what they had expected to find. This is an indication that maybe their participants were even an exception within the group of vocationally educated people and might not even represent this group perfectly. So it is not possible to generalize from this specific group to a larger majority.

15. Discussion; 5.4; end of paragraph 3. The authors state that KiesKompas was struggling to reach the required number of participants. Any suggestions why this was the case? This was confusing to me because in the Methods section the authors claimed that the right number of participants was found in the end. How did KiesKompas end up collecting the right amount of data in the end and how did they contact and reach participants in the first place? Please add a short explanation of how this collection service works.

Reviewer #2: This is an investigation amongst vocationally trained young adults and tests whether misleading graphs can give readers a distorted view of the underlying data.

This is an interesting study that seems unlikely to be done, especially because the main question seems to be so self-evident. If one gets access to a set of data via some graphical illustration, then it seems obvious that the graphs being misleading also leads to a distorted view of the data. Hence, it would not be worth simply investigating such an obvious question. However, the authors go one step further and then ask, how graph correction could be most effectively done to mitigate this distortion effect. As such, there are valuable practical implications of this study that are well worth publishing.

There are some methodological questions that should be explained in the manuscript with stronger argumentation: (i) Why exactly was this population selected for the present study? After all, misleading graphs are a problem for all of us. (ii) What is the benefit of the think-aloud task over other qualitative methods that might also have been helpful? (iii) It seems not surprising that correcting misleading graphs eventually were shown to be less misleading, and also that there might be a learning curve at hand. Given the intuition that this “is not news”, how are these results relevant vis-à-vis both this basic intuition and the existing literature? In other words, what makes these findings so relevant, especially since there was no difference between the specific type of the correction? I believe further strengthening on these arguments might help strengthen the case for this project, since it is not so clear otherwise why it should be relevant given the unsurprising nature of the results. After all, one could argue, that “everybody knows that we can benefit from correction of misleading graphs, not just young adults, and there is no value in a study for this”. This does not say that the study is not needed, but that it is important to make a clear case right from the beginning up until the end of why this study is necessary and relevant.

I am not aware how this pertains to the journal’s guidelines, but I would suggest to put the links in the methods section either in the footnotes or as a regular reference.

Please check how a study us usually outlined: You described the hypotheses in the “analysis” chapter. This is too late. The goals, research questions and hypotheses need to be elucidated right at the beginning (or, somewhere in the introductory chapter). Introducing them in the methods, analysis or discussion chapter would be too late, since it must tether the whole project to what one actually wants to investigate.

Since the paper discusses misleading graphs, it would be helpful to readers to actually see some illustrations of what you mean by misleading graphs and correct graphs. Likewise, it would be helpful to have some visualization of the study design.

Overall, this is certainly an intriguing project and I thank the authors for their research.

Reviewer #3: Please see the attached file with the full comment.

Paper summary

This paper presents an experimental study of graph reading and interpretation in the niche area of vocationally educated students. The methodology involves a mixed methods design with interviews and a survey. Despite a well justified and numerous sample, the authors did not present their findings and discuss their insights well.

Positive Aspects

The research addresses an interesting and relevant topic with practical implications and does so through a compelling combination of qualitative and quantitative methods. The discussion highlights some valuable insights and these aspects would deserve more space in terms of practical implications and applications. In addition, the scope and choice of targets show sensitivity to the social relevance of the issue and the impact of the findings.

Limitations

The reviewed paper suffers from lack of precision and organization in exposition, not proper references and context for strong claims. Unclear presentation of results and missing details hinder readability and comprehension of the study conducted.

**Do you want your identity to be public for this peer review?** For information about this choice, including consent withdrawal, please see our Privacy Policy

Reviewer #1: **Yes:** Edith Haim

Reviewer #2: No

Reviewer #3: No

---

## [Author Response · Author response to Decision Letter 1]

5 Mar 2025

Please see Response to Reviewers document and Cover Letter.

---

## [Decision Letter · Decision Letter 1]

8 Sep 2025

Dear Dr. Wijnker,

Thank you for submitting your manuscript to PLOS ONE. After careful consideration, we feel that it has merit but does not fully meet PLOS ONE’s publication criteria as it currently stands. Therefore, we invite you to submit a revised version of the manuscript that addresses the points raised during the review process.

**ACADEMIC EDITOR:**

Dear Authors,

Thank you for submitting the revised version of your manuscript to PLOS ONE. The manuscript has now been re-evaluated by two reviewers.

While one reviewer is satisfied with the revisions, the other raises significant concerns regarding readability and structure. After carefully reviewing the manuscript and the reports, I agree that although the study addresses an important and timely topic, the presentation requires substantial revision before the paper can be considered further.

In its current form, the manuscript is very lengthy, fragmented into many small subsections, and difficult to follow as a coherent narrative. The Introduction and Discussion in particular could be shortened considerably without loss of substance. This would help ensure that the main contribution of the study is clear and accessible for the general scientific readership.

I therefore ask that you:

Substantially condense the manuscript, especially the Introduction and Discussion. Aim to reduce redundancy and remove details that can be streamlined or moved to supplementary material.Reconsider the structure, merging smaller subsections into larger, more integrated sections to improve readability.Sharpen the central message, focusing on the key contribution that misleading graphs can distort perception and that corrective exposure—regardless of design—can mitigate these effects.Present limitations more concisely, for example regarding the challenges of using visual analogue scales, which currently receive extended and somewhat repetitive treatment.

Please prepare a carefully revised manuscript that addresses these points, and include a detailed response letter describing how you have responded to each reviewer and editorial comment.

I look forward to receiving your revised submission.

We look forward to receiving your revised manuscript.

Kind regards,

Nicola Diviani

Academic Editor

PLOS ONE

Journal Requirements:

Reviewers' comments:

Reviewer's Responses to Questions

**Comments to the Author**

Reviewer #1: All comments have been addressed

Reviewer #4: (No Response)

2. Is the manuscript technically sound, and do the data support the conclusions?

Reviewer #1: Yes

Reviewer #4: Yes

3. Has the statistical analysis been performed appropriately and rigorously?

Reviewer #1: Yes

Reviewer #4: Yes

4. Have the authors made all data underlying the findings in their manuscript fully available?

Reviewer #1: Yes

Reviewer #4: Yes

5. Is the manuscript presented in an intelligible fashion and written in standard English?

Reviewer #1: Yes

Reviewer #4: No

Reviewer #1: The authors have addressed all of my comments and improved the manuscript substantially. I consider it now suitable for publication.

Reviewer #4: Whilst this is an interesting study, it is, at least for this reviewer, unreadable in its present format. It is lengthy, broken up into small sections so that it is hard to get a feel for the underlying study, and demonstrates the considerable challenges associated with analysing Visual analogue scales. I would suggest the authors re-examine the text, merge many of the sub- sections into larger textual sections, and generally try to think about the reader experience in a text that is much briefer. I suspect the salient point that its easy to mislead people with deceptive graphs is worth saying, and that some corrective exposure to the issues is worthwhile is also worth saying, but at much shorter length

**Do you want your identity to be public for this peer review?** For information about this choice, including consent withdrawal, please see our Privacy Policy

Reviewer #1: No

Reviewer #4: No

---

## [Editor Report · Decision Letter 2]

16 Dec 2025

Debunking misleading graphs effectively:

How vocationally educated young adults perceive graphs

PONE-D-24-25302R2

Dear Dr. Wijnker,

We’re pleased to inform you that your manuscript has been judged scientifically suitable for publication and will be formally accepted for publication once it meets all outstanding technical requirements.

Kind regards,

Nicola Diviani

Academic Editor

PLOS One
---

## [Editor Report · Acceptance letter]

PONE-D-24-25302R2

PLOS One

Dear Dr. Willems,

I'm pleased to inform you that your manuscript has been deemed suitable for publication in PLOS One. Congratulations! Your manuscript is now being handed over to our production team.

Kind regards,

on behalf of

Dr. Nicola Diviani

Academic Editor

PLOS One